# High-Sensitivity Troponin I and Creatinine Kinase-Myocardial Band in Screening for Myocardial Injury in Patients with Carbon Monoxide Poisoning

**DOI:** 10.3390/diagnostics10040242

**Published:** 2020-04-21

**Authors:** June-Sung Kim, Byuk Sung Ko, Chang Hwan Sohn, Youn-Jung Kim, Won Young Kim

**Affiliations:** 1Asan Medical Center, Department of Emergency Medicine, College of Medicine, University of Ulsan, Seoul 05505, Korea; 2Department of Emergency Medicine, College of Medicine, University of Hanyang, Seoul 04763, Korea

**Keywords:** cardiomyopathy, carbon monoxide, biomarkers

## Abstract

Myocardial dysfunction due to acute carbon monoxide (CO) poisoning is common and associated with poor outcomes. The role of cardiac markers, including creatine kinase-myocardial band (CK-MB), high-sensitivity troponin I (hsTnI), and brain natriuretic peptide (BNP), in identifying patients with CO-induced cardiomyopathy were evaluated. This single-center, retrospective cohort study included 905 consecutive adult patients in the CO poisoning registry from February 2009 to December 2019. Cardiomyopathy was defined as any abnormality on transthoracic echocardiography (TTE), including left ventricular systolic and diastolic dysfunction, right ventricular dysfunction, and wall motion abnormalities. The areas under receiver operating curves (AUCs) for biomarkers were compared. Of the 850 included patients, 101 (11.9%) had CO-induced cardiomyopathy. Initial and peak hsTnI and CK-MB concentrations, and initial BNP concentrations were significantly higher in patients with than without cardiomyopathy (all *P*-values < 0.01), but the AUCs were higher for hsTnI (0.894) and CK-MB (0.864) than for BNP (0.796). Initial TnI > 0.01 ng/mL and CK-MB > 1.5 ng/mL each had 95% sensitivity and 97% negative predictive value for CO-induced cardiomyopathy. Higher hsTnI or CK-MB levels on admission can identify patients at high-risk of CO-induced cardiomyopathy and can be a screening tool for CO poisoning.

## 1. Introduction

Acute carbon monoxide (CO) poisoning is a common cause of emergency department (ED) visits worldwide [1]. CO competes with oxygen in binding to hemoglobin and causes tissue hypoxia by reducing oxygen delivery to tissues [2]. This toxic effect frequently affects the heart, resulting in the development of cardiomyopathy, angina attacks, heart failure, cardiogenic shock, and cardiac arrest [3]. Although the exact pathophysiology of CO-induced cardiomyopathy has not been determined, cardiomyopathy has been associated with poor outcomes [4]. Therefore, early assessment and proper management of cardiac dysfunctions after CO poisoning may improve patient outcomes [5]. Although echocardiography is regarded as the gold standard for diagnosing cardiomyopathy, echocardiographic findings associated with cardiomyopathy remain unclear [6]. Moreover, the limited availability and high costs of echocardiography make this method impractical for evaluating all patients with CO intoxication.

The creatine kinase-myocardial band (CK-MB), troponin I (TnI), and brain natriuretic peptide (BNP) are cardiac proteins expressed in heart muscle and are elevated following myocardial damage [7,8,9]. Although these enzymes are also elevated in patients with CO-induced cardiomyopathy [10,11], few studies to date have analyzed the direct correlation between these markers and CO-induced cardiomyopathy on transthoracic echocardiography (TTE). Thus, the direct relationships between elevated concentrations of each cardiac enzyme and abnormalities on TTE remain unknown.

We hypothesized that concentrations of cardiac biomarkers at admission to the ED and at hospital admission may reflect CO-associated cardiac injury and may be complementary to TTE-derived parameters. The present study therefore measured initial and peak concentrations of these cardiac enzymes in a prospective registry of patients brought to the ED for CO poisoning to assess whether these enzymes could identify patients with CO-induced cardiomyopathy.

## 2. Materials and Methods

### 2.1. Study Setting

This registry-based, single-center, retrospective observational study was performed on patients aged ≥ 18 years brought to the ED of Asan Medical Center, a tertiary referral academic center in Seoul, Korea, with 2800 beds and 130,000 annual visits, from 1 February 2009 to 31 December 2016. Data on all consecutive adult patients aged ≥ 18 years were prospectively collected in the registry [12]. This study was approved by the Institutional Review Board of the Asan Medical Center (Study No. 2020-0457), which waived the requirement for informed consent because of the retrospective characteristics of the study.

### 2.2. Study Population and Definition of Variables

A diagnosis of acute CO poisoning was based on history-taking by the emergency physicians on duty and a carboxyhemoglobin (CO-Hb) concentration > 5% (> 10% in current smokers) [13]. Patients were excluded if they had fire-related CO poisoning, were readmitted, or had TTE abnormalities within one year. Patients were also excluded if TTE was not performed, cardiac markers were not measured, or if their visit was due to delayed presentation. All patients had been treated with oxygen supplied via a face mask with a reservoir bag and hyperbaric oxygen therapy, followed by standard treatments [14].

Clinical data were extracted from the electronic medical records of registry-enrolled patients. Factors recorded included age, gender, intentionality of CO poisoning, co-ingestion of alcohol, co-ingestion of drugs, duration of CO exposure, past medical histories, and current smoking. The duration of CO exposure was reported by the patients or their guardians, or by the emergency medical personnel. All patients underwent routine laboratory tests on arrival at the ED, including measurements of CO-hemoglobin, lactate, and creatinine concentrations.

### 2.3. Cardiac Enzymes and TTE Parameters

Cardiac enzymes, including CK-MB, hsTnI, and BNP, were routinely measured on arrival at the ED. CK-MB and hsTnI were again measured 2–4 h later and once per day during admission, and peak concentrations were recorded. Concentrations of hsTnI were measured with a three-site sandwich immunoassay using TnI-Ultra direct chemiluminometric technology (ADVI Centaur XPT; Siemens, Munich, Germany), with the cutoff value set at 0.04 ng/mL, the 99th percentile in the normal population. CK-MB and BNP concentrations were measured using a two-site sandwich immunoassay with direct chemiluminometric technology (ADVI Centaur XPT; Siemens, Munich, Germany). The cutoff value for CK-MB was set at 5.0 ng/mL, whereas the cutoff value for BNP was dependent on age and gender (e.g., 39 pg/mL for men aged 55–64 years) [15]. Patients with normal range of all cardiac markers while in the ED or hospital were defined as not having cardiomyopathy because the development of CO-induced cardiomyopathy without elevation of cardiac biomarkers is uncommon.

CO-induced cardiomyopathy was diagnosed via two-dimensional TTE performed by experienced sonographers or cardiologists. Cardiac dysfunction was defined as any indication of left ventricle (LV) systolic, LV diastolic, or right ventricle (RV) dysfunction, and/or wall motion abnormalities. Echocardiographic parameters were determined according to the guidelines of the American Society of Echocardiograph [16]. LV ejection fraction was calculated using the Teichholz method via M-mode of the parasternal long-axis, with a value < 50% considered indicative of LV systolic dysfunction. E/e’ ratio was measured using tissue Doppler imaging in both the lateral and septal mitral origins on a four-chamber view, with a ratio > 15 defined as indicating LV diastolic dysfunction. RV dysfunction was defined as visually determined reduced contractility or dilation. All echocardiographic results were reviewed by three independent board-certified emergency medicine physicians (J.S.K., Y.J.K., and C.H.S.).

The primary objective of this study was to evaluate the diagnostic abilities of initial and peak CK-MB and hsTnI concentrations, and initial BNP concentrations, during ED stay or hospital admission to predict CO-induced cardiomyopathy.

### 2.4. Statistical Analysis

Continuous variables are reported as median and interquartile range (IQR) due to their non-normal distribution, as determined by the Kolmogorov–Smirnov test, and were compared using the Mann–Whitney U-test. Categorical variables are expressed as frequency and percentages, and compared using the Chi-square test or Fisher’s exact test, as appropriate. Receiver operating characteristic (ROC) curves were determined and areas under ROC curves (AUCs) were calculated to compare the diagnostic abilities of cardiac markers. The optimal cutoff values of these biomarkers were determined using the Youden index (sensitivity + specificity − 1), followed by the calculation of sensitivity and specificity with standard statistical methods. Two-tailed *P*-values < 0.05 were considered statistically significant. All statistical analyses were performed using SPSS Statistics for Windows version 23.0 (IBM Corp., Armonk, NY, USA).

## 3. Results

### 3.1. Study Population

Figure 1 shows the flowchart of the included population. During the study period, 905 adult patients were included in the CO registry at the study facility. Of these, 55 patients were excluded, including 5 who were readmitted, 4 in whom cardiac markers were not measured, 19 with delayed presentation, 11 with fire-related CO intoxication, 4 with drug intoxication, and 12 who lacked TTE data or had previously reported TTE abnormalities. Of the 850 included patients, 101 (11.9%) had myocardial dysfunction on TTE and 749 (88.1%) did not.

### 3.2. Baseline Patient Characteristics

Table 1 presents the baseline characteristics of patients with CO poisoning. The median age was 41.0 years with male predominance in both patient groups, those without and with cardiomyopathy. Rates of intentionality, alcohol co-ingestion, drug co-ingestion, and current smoking did not differ significantly between the two groups, whereas both hypertension and diabetes mellitus were more frequent in patients with than without cardiomyopathy. Initial creatinine concentration was slightly higher in patients with than without cardiomyopathy (1.0 mg/dL vs. 0.9 mg/dL), although the proportions of patients with chronic kidney disease were similar between the two groups. LV ejection fraction was significantly lower (52.0% vs. 61.0%), and E/e’ higher (9.0 vs. 8.0) in the cardiomyopathy group.

### 3.3. Frequency of Various Cardiac Dysfunctions

The frequencies of various types of myocardial dysfunction in patients with cardiomyopathy are shown in Table 2. LV systolic dysfunction was the most common (74.3%), followed by wall motion abnormalities (61.4%). LV diastolic and RV dysfunction were relatively uncommon.

### 3.4. Comparisons of Cardiac Biomarkers

Table 3 shows initial and peak concentrations of cardiac markers in patients with and without cardiomyopathy. Initial median CK-MB (25.60 pg/mL vs. 1.55 ng/mL), hsTnI (1.31 pg/mL vs. 0.01 ng/mL), and BNP (89.00 pg/mL vs. 15.00 pg/mL) concentrations were significantly higher in the cardiomyopathy group. Peak CK-MB (31.80 pg/mL vs. 1.80 ng/mL) and hsTnI (1.84 pg/mL vs. 0.02 ng/mL) concentrations were higher in patients with than without myocardial dysfunction. Moreover, all initial and peak concentrations of CK-MB, hsTnI, and BNP were significantly higher in each type of dysfunction, including LV, RV dysfunction, and abnormal wall motions (Appendix A).

### 3.5. Diagnostic Performances of Cardiac Biomarkers as Predictors of CO-Induced Cardiomyopathy

Figure 2 shows the ROC curves of cardiac markers for predicting CO-induced cardiomyopathy. The AUC was highest for peak hsTnI (0.898), but was also higher for initial hsTnI (0.894), peak CK-MB (0.877), and initial CK-MB (0.874). BNP had the smallest AUC (0.796). Comparison of AUC between initial and peak concentrations of each biomarker showed no statistical differences (difference between areas 0.003; *p* = 0.20 for initial vs. peak CK-MB, difference between areas 0.004; *p* = 0.71 for initial vs. peak hsTnI) Using the ROC curves, we found that the optimal cutoff values were 1.5 ng/mL for initial CK-MB, 1.9 ng/mL for peak CK-MB, 0.01 ng/mL for initial hsTnI, 0.03 ng/mL for peak hsTnI, and 14.5 ng/mL for BNP. Above their cutoffs, both initial CK-MB and hsTnI had high sensitivities (95.1% and 96.0%, respectively) and negative predictive values (97.0% and 97.6%, respectively) (Table 4). Meanwhile, combinations of the cutoff values could not improve diagnostic performances.

## 4. Discussion

The present study found that CO-induced cardiomyopathy developed in about one out of eight patients with CO poisoning. All CK-MB, hsTnI, and BNP concentrations at initial presentation were associated with CO-induced TTE-determined myocardial dysfunction. The diagnostic abilities of CK-MB and hsTnI levels were better than that of BNP. Although the serial measurements of CK-MB and hsTnI concentrations identified their peak levels, these peak concentrations were not superior to concentrations at admission for predicting CO-induced cardiomyopathy.

The heart is the primary target of acute CO poisoning via direct toxic effect or secondary to ischemic hypoxia [2,3]. Previous reports announced that CO caused various cardiovascular complications, such as arrhythmia, heart failure, and myocardial infarction [17]. Although its exact epidemiology and pathophysiology have not been fully discovered, recent studies tried to evaluate the risk factors, patterns, clinical outcomes, and prognosis [18,19]. Jung et al. reported that CO-induced cardiomyopathy had similar characteristics and patterns with stress-induced cardiomyopathy with favorable outcomes. Moreover, they suggested that myocardial stunning caused by a catecholamine surge plays a key mechanism in the occurrence of cardiomyopathy [19].

The incidence of cardiomyopathy induced by CO intoxication has been reported to range from 10% to 70% [18,19], with differences due mainly to included populations and different definitions. A recent prospective study reported that the incidence of myocardial dysfunction was as high as 75% (32/43) [20]. That study, however, defined CO-induced cardiomyopathy as abnormal systolic dysfunction (LV EF < 50%) on TTE and only included patients with hsTnI concentrations > 0.046 ng/mL. Another retrospective analysis, which reported myocardial injury in 37% (85/230) of patients, defined cardiomyopathy as TnI ≥ 0.7 ng/mL or CK-MB ≥ 5.0 ng/mL without TTE data, and analyzed only those patients with moderate to severe CO intoxication [4]. Our study defined CO-induced cardiomyopathy not only by LV systolic dysfunction, but also by LV diastolic and RV dysfunction and wall motion abnormalities without LV or RV dysfunctions. Most toxin-induced myocardial dysfunctions likely involve both the RV and LV. Moreover, LV diastolic dysfunction or wall motion abnormality without decreased systolic function may be a mild, early stage of cardiomyopathy [21]. Future studies are needed to analyze the clinical impact of each type of myocardial dysfunction on patients with CO intoxication.

It remains unclear whether myocardial injury leads to releases of cardiac enzymes. The possible pathophysiology of cardiomyopathy may involve the activation by CO-induced tissue hypoxia of nitrogen oxide-mediated cardiomyocyte apoptosis [22]. Moreover, CO-hemoglobin may have a direct toxic effect on the mitochondria, rapidly reducing myocardial oxygen reserve [23,24]. These impairments in energy metabolism may induce myocardial fiber necrosis and release components of the contractile apparatus, such as TnI, CK-MB, and BNP [25,26]. However, little is known about the ability of each biomarker to recognize early-phase myocardial dysfunction. Moreover, studies to date have yielded inconsistent results because of differences in biomarker types (troponin I or troponin T), cutoff values, severity of illnesses, time to measurement, and diagnostic criteria of cardiomyopathy. Our findings demonstrated that all initial and peak concentrations of cardiac enzymes were higher in patients with than without cardiomyopathy. Moreover, CK-MB and hsTnI concentrations had excellent diagnostic abilities to detect TTE-assessed cardiomyopathies with high sensitivity. Higher hsTnI or CK-MB levels on ED admission may be able to identify patients at high-risk of CO-induced cardiomyopathy, suggesting that these markers may constitute a screening tool to determine whether patients with acute CO poisoning should be assessed by TTE.

Measurements of the optimal cutoffs for initial hsTnI and CK-MB concentrations by the Youden index method showed that these cutoff concentrations in patients with CO-induced cardiomyopathy were 0.01 ng/mL and 1.5 ng/mL, respectively, lower than those previously determined in patients with acute coronary syndrome (0.04 ng/mL and 5.0 ng/mL, respectively). These findings indicate that even low levels of cardiac enzymes can be diagnostic of myocardial injury, and that only those patients with initial and peak levels above cutoff concentrations require further evaluation, such as echocardiography. Because the concentrations of these enzymes increase at different rates, we evaluated whether combinations of their cutoff values could enhance their diagnostic abilities, including sensitivity and specificity. However, none of these combinations improved outcomes.

This study had several limitations. First, because it was a retrospective cohort study of patients enrolled in a single-center registry, it is difficult to generalize from these results and adapt them to other environments. Second, the times between the measurement of cardiac markers and echocardiography could not be controlled. However, 95% of patients underwent echocardiography within 48 h of admission. Third, baseline TTE parameters and cardiac biomarker levels prior to ED arrival were not measured in all patients. Thus, we could not determine the true incidence of CO-induced cardiomyopathy. Finally, this registry did not include data about clinical outcomes, such as recovery rate, mortality, and neurologic deficit, and could not provide clinical impacts of CO-induced cardiomyopathy. A well-designed prospective study will be needed to reveal these questions.

## 5. Conclusions

Both elevated CK-MB and hsTnI concentrations at ED presentation are reliable biomarkers of CO-induced cardiomyopathy on echocardiography. Serial measurements of cardiac enzymes do not increase their ability to diagnose cardiomyopathy. Cardiac enzymes, excluding BNP, may serve as promising screening tools to detect myocardial dysfunction in patients with CO poisoning.

## Figures and Tables

**Figure 1 diagnostics-10-00242-f001:**
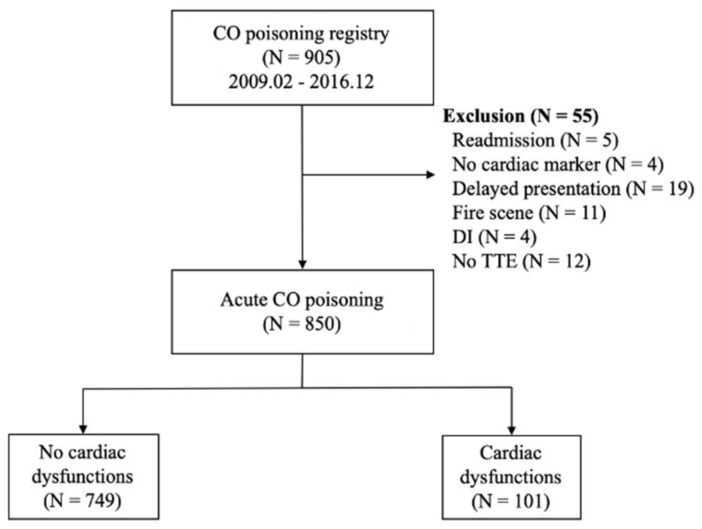
Flow diagram for the study population. Abbreviations: CO = carbon monoxide; DI = drug intoxication.

**Figure 2 diagnostics-10-00242-f002:**
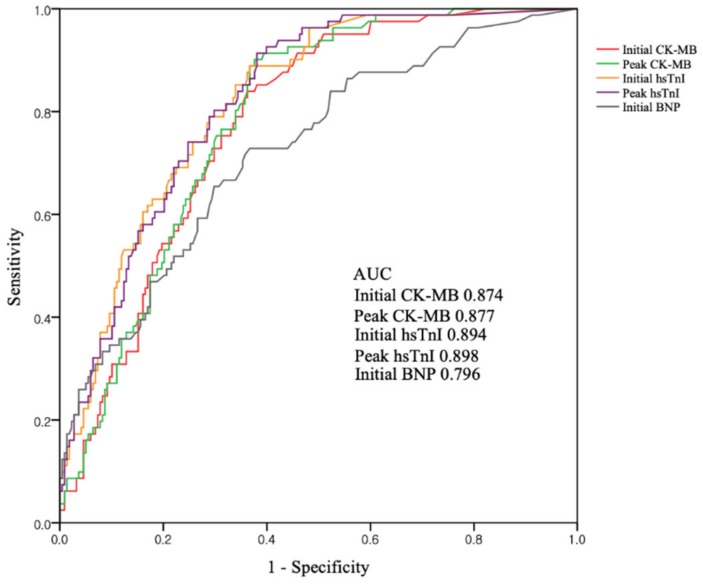
ROC curves for cardiac biomarkers as predictors of CO-induced cardiomyopathy. Abbreviations: AUC = area under the curve; CO = carbon monoxide; CK-MB = creatinine kinase-myocardial band; hsTnI = high-sensitivity troponin I; BNP = brain natriuretic peptide.

**Table 1 diagnostics-10-00242-t001:** Baseline characteristics of patients with CO-induced cardiomyopathy on TTE.

Characteristics	Total (*n* = 850)	No Cardiomyopathy (*n* = 749)	Cardiomyopathy (*n* = 101)	*p*
Age (year)	41.0 (32.0–53.0)	34.0 (27.0–46.5)	44.0 (34.5–64.5)	0.25
Male	301 (64.7)	243 (66.8)	58 (57.4)	0.09
Intentional Poisoning	148 (31.9)	120 (33.1)	28 (27.7)	0.31
Alcohol Co-ingestion	200 (44.1)	153 (43.3)	47 (46.5)	0.57
Drug Co-ingestion	67 (14.8)	53 (15.0)	14 (13.9)	0.77
CO Exposure Time (hour)	4.0 (2.0–7.5)	4.0 (2.0–8.0)	7.0 (4.5–11.0)	0.03
Past Illness				
HTN	79 (17.4)	53 (15.0)	26 (25.7)	0.01
DM	43 (9.5)	27 (7.6)	16 (15.8)	0.01
CKD	3 (0.7)	3 (0.8)	0 (0.0)	0.35
Current Smoker	59 (13.0)	45 (12.7)	14 (13.9)	0.77
Laboratory				
Initial CO-HB	25.6 (11.1–37.2)	38.2 (25.3–48.9)	34.8 (27.0–42.2)	0.17
(%)				
Initial Lactate	2.1 (1.1–3.7)	3.4 (1.3–6.5)	3.0 (2.1–6.0)	< 0.01
(mmol/L)				
Creatinine	0.8 (0.7–1.0)	0.9 (0.6–1.1)	1.0 (0.7–1.2)	< 0.01
(mg/dL)				
TTE Parameters				
LVEF (%)	59.0 (51.0–62.3)	61.0 (59.0–65.0)	52.0 (45.5–55.0)	< 0.01
E/e’	9.0 (7.0–12.0)	8.0 (6.5–9.0)	9.0 (7.5–12.5)	0.01

Data are presented as *N* (%) or median with interquartile ranges. Abbreviations: CO = carbon monoxide; SCMP = stress-induced cardiomyopathy; TTE = transthoracic echocardiography; ED = emergency department; HTN = hypertension; DM = diabetes mellitus; EKG = electrocardiogram; STTC = ST segment and T wave change; Hb = hemoglobin.

**Table 2 diagnostics-10-00242-t002:** Type of CO-induced myocardial dysfunctions on TTE.

Type of Cardiac Dysfunction	Frequency, *n* (%)
LV dysfunction	
Systolic dysfunction	75 (74.3)
Diastolic dysfunction	28 (27.7)
RV dysfunction	21 (20.8)
Wall motion abnormalities	62 (61.4)

Abbreviations: TTE = transthoracic echocardiography; LV = left ventricle; RV = right ventricle.

**Table 3 diagnostics-10-00242-t003:** Ability of cardiac biomarkers to predict CO-induced cardiomyopathy.

Cardiac Markers	Total (*n* = 850)	No Cardiomyopathy (*n* = 749)	Cardiomyopathy (*n* = 101)	*p*
Initial CK-MB	2.00 (0.70–9.80)	1.55 (0.60–4.95)	25.60 (9.20–60.30)	<0.01
(ng/mL)				
Peak CK-MB	2.50 (0.80–13.30)	1.80 (0.65–5.70)	31.80 (13.80–73.60)	<0.01
(ng/mL)				
Initial hsTnI	0.03 (0.01–0.27)	0.01 (0.01–0.12)	1.31 (0.38–3.16)	<0.01
(ng/mL)				
Peak hsTnI	0.05 (0.01–0.50)	0.02 (0.01–0.23)	1.84 (0.58–4.90)	<0.01
(ng/mL)				
BNP	15.00 (5.00–41.50)	15.00 (5.00–41.50)	89.00 (29.00–255.00)	<0.01
(pg/mL)				

Abbreviations: CO = carbon monoxide; CK-MB = creatinine kinase-myocardial band; hsTnI = high-sensitivity troponin I; BNP = brain natriuretic peptide.

**Table 4 diagnostics-10-00242-t004:** Performance parameters for predictors of CO-induced cardiomyopathy.

Variables	Sensitivity (%)	Specificity (%)	PPV (%)	NPV (%)	PLR	NLR
Initial CK-MB > 1.5 ng/mL	95.1	44.0	32.0	97.0	1.7	1.1
Peak CK-MB > 1.9 ng/mL	97.0	46.2	33.3	98.3	1.8	0.1
Initial hsTnI > 0.01 ng/mL	96.0	41.8	31.4	97.4	1.7	0.1
Peak hsTnI > 0.03 ng/mL	95.2	44.0	32.5	97.0	1.7	0.1
BNP > 14.5 pg/mL	87.7	37.9	33.5	89.6	1.4	0.3
Initial CK-MB > 1.5 + hsTnI > 0.01	96.0	31.0	27.9	96.6	1.4	0.1
Peak CK-MB > 1.9 + hsTnI > 0.03	97.0	36.3	29.7	97.8	1.5	0.1

Abbreviations: CO = carbon monoxide; PPV = positive predictive value; NPV = negative predictive value; PLR = positive likelihood ratio; NLR = negative likelihood ratio; CK-MB = creatinine kinase-myocardial band; hsTnI = high-sensitivity troponin I; BNP = brain natriuretic peptide.

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
