# Peer review of "High-Sensitivity Troponin I and Creatinine Kinase-Myocardial Band in Screening for Myocardial Injury in Patients with Carbon Monoxide Poisoning"

_diagnostics, 2020, doi:10.3390/diagnostics10040242_

Round 1

Reviewer 1 Report

It is my pleasure to review this interesting work about Carbon Monoxide Poisoning related cardiomyopathy. Initial troponin and CK-MB levels could predict this problem from a long-term cohort study. BNP seems not a good tool for this problem.

May the author provide the level of the cardiac biomarker in the groups with different dysfunction (such as LV dysfunction VS. RV dysfunction or with abnormal wall motion VS. without abnormal wall motion)?

Could the author provide clinical outcomes about the different levels of the cardiac biomarkers?

Author Response

It is my pleasure to review this interesting work about Carbon Monoxide Poisoning related cardiomyopathy. Initial troponin and CK-MB levels could predict this problem from a long-term cohort study. BNP seems not a good tool for this problem.

May the author provide the level of the cardiac biomarker in the groups with different dysfunction (such as LV dysfunction VS. RV dysfunction or with abnormal wall motion VS. without abnormal wall motion)?

Response> Thank you for your great suggestion. We performed additional analysis as your suggestions. All initial and peak concentrations of CK-MB, hsTnI, and BNP were significantly higher in each dysfunction, including LV, RV dysfunction, and abnormal wall motions. We provided these tables in supplement tables and mentioned in the results section.

Supplement Table 1. Ability of cardiac biomarkers to predict CO-induced LV dysfunction.

Cardiac markers

Total

(N = 850)

No LV dysfunction

(N = 761)

LV dysfunction

(N = 89)

P

Initial CK-MB

(ng/mL)

1.50 (0.60–5.70)

1.60 (0.60–5.20)

24.2 (9.4–60.3)

< 0.01

Peak CK-MB

(ng/mL)

1.80 (0.60–8.65)

1.90 (0.70–6.00)

31.35 (13.80–73.60)

< 0.01

Initial hsTnI

(ng/mL)

0.01 (0.01–0.16)

0.02 (0.01–0.13)

1.41 (0.38–3.16)

< 0.01

Peak hsTnI

(ng/mL)

0.03 (0.01–0.39)

0.03 (0.01–0.27)

1.82 (0.67–4.90)

< 0.01

BNP

(pg/mL)

18.00 (6.0–56.00)

15.00 (6.00–43.00)

78.50 (28.00–242.00)

< 0.01

Abbreviations: CO = carbon monoxide; LV = left ventricle; CK-MB = creatinine kinase-myocardial band; hsTnI = high-sensitivity troponin I; BNP = brain natriuretic peptide.

Supplement Table 2. Ability of cardiac biomarkers to predict CO-induced RV dysfunction.

Cardiac markers

Total

(N = 850)

No RV dysfunction

(N = 829)

RV dysfunction

(N = 21)

P

Initial CK-MB

(ng/mL)

1.50 (0.60–5.70)

1.80 (0.60–7.90)

31.80 (13.05–60.25)

< 0.01

Peak CK-MB

(ng/mL)

1.80 (0.60–8.65)

2.30 (0.70–11.10)

35.40 (20.90–89.20)

< 0.01

Initial hsTnI

(ng/mL)

0.01 (0.01–0.16)

0.03 (0.01–0.24)

2.36 (0.56–4.64)

< 0.01

Peak hsTnI

(ng/mL)

0.03 (0.01–0.39)

0.04 (0.01–0.42)

2.36 (0.68–8.88)

< 0.01

BNP

(pg/mL)

18.00 (6.00–56.00)

17.00 (6.00–53.00)

112.00 (48.50–306.00)

< 0.01

Abbreviations: CO = carbon monoxide; RV = right ventricle; CK-MB = creatinine kinase-myocardial band; hsTnI = high-sensitivity troponin I; BNP = brain natriuretic peptide.

Supplement Table 3. Ability of cardiac biomarkers to predict CO-induced WMA.

Cardiac markers

Total

(N = 850)

No WMA

(N = 788)

WMA

(N = 62)

P

Initial CK-MB

(ng/mL)

1.50 (0.60–5.70)

1.70 (0.60–5.70)

29.70 (10.20–76.70)

< 0.01

Peak CK-MB

(ng/mL)

1.80 (0.60–8.65)

2.10 (0.70–8.05)

33.90 (20.700–97.90)

< 0.01

Initial hsTnI

(ng/mL)

0.01 (0.01–0.16)

0.02 (0.01–0.16)

1.63 (0.45–3.26)

< 0.01

Peak hsTnI

(ng/mL)

0.03 (0.01–0.39)

0.03 (0.01–0.32)

1.95 (0.69–5.94)

< 0.01

BNP

(pg/mL)

18.00 (6.00–56.00)

16.00 (6.00–46.50)

114.50 (53.00–294.00)

< 0.01

Abbreviations: CO = carbon monoxide; WMA = wall motion abnormality; CK-MB = creatinine kinase-myocardial band; hsTnI = high-sensitivity troponin I; BNP = brain natriuretic peptide.

Could the author provide clinical outcomes about the different levels of the cardiac biomarkers?

Response> We agreed with your opinion that whether cardiac biomarkers can predict not only the development of cardiomyopathy but also clinical outcomes including mortality, short, and long-term neurologic complications. Previous reports (Jung et al. Circ J 2014, 78, 1437–1444, Cha et al. Clin Toxicol 2016, 54, 481–487) announced that most cardiomyopathies were resolved within 4 to 6 weeks, and showed favorable outcomes. Regretfully, we conducted a retrospective registry-based study and did not have additional data of such clinical outcomes. We have the plan to evaluate the exact clinical outcomes of CO-induced cardiomyopathy via a well-designed prospective study in the future.

Reviewer 2 Report

The manuscript entitled “High-sensitivity troponin I and creatinine kinase-myocardial band in screening for myocardial injury in patients with carbon monoxide poisoning” represents a single-center, retrospective cohort study evaluating the role of cardiac biomarkers in identifying CO-induced cardiomyopathy assess by TTE.

My comments are as follows:

  • Please recheck the numbers in Table 3: initial CK-MB, peak CK-MB, initial hsTnI, and peak hsTnI in the Total group). These values should fall within the range between No cardiomyopathy and Cardiomyopathy group.

  • Discussion: “Both CK-MB and hsTnI concentrations at initial presentation, but not BNP concentration, were associated with CO-induced TTE-determined myocardial dysfunction.” Although the performance of BNP may not be as good as other markers in the present study, BNP levels were greater in Cardiomyopathy group than the No cardiomyopathy group (Table 2; P<0.01). Please consider rewording this statement.

  • Discussion: “Although serial measurements of CK-MB and hsTnI concentrations identified their peak levels, these peak concentrations were not superior to concentrations at admission for predicting CO-induced cardiomyopathy.” While statement may be true, it is encouraged to add the comparison of AUCs to the results demonstrating statistical significance.

  • Whether CO-induced cardiomyopathy is a clinical entity has yet to be established. This should be addressed by discussing the current knowledge from preliminary studies.

  • Discussion: “Because the concentrations of these enzymes increase at different rates, we evaluated whether combinations of their cutoff values could enhance their diagnostic abilities, including sensitivity and specificity. However, none of these combinations improved outcomes.” Relevant data should be reported in the Results section.

Author Response

The manuscript entitled “High-sensitivity troponin I and creatinine kinase-myocardial band in screening for myocardial injury in patients with carbon monoxide poisoning” represents a single-center, retrospective cohort study evaluating the role of cardiac biomarkers in identifying CO-induced cardiomyopathy assess by TTE.

My comments are as follows:

Please recheck the numbers in Table 3: initial CK-MB, peak CK-MB, initial hsTnI, and peak hsTnI in the Total group). These values should fall within the range between No cardiomyopathy and Cardiomyopathy group.

Response> We are sorry to make you confusing. We analyzed the data again and found that the numbers in Table 3 were typos, and those numbers were not 25, 75 quantiles. We corrected the numbers in Table 3 and the main text in the Result section.

Table 3. Ability of cardiac biomarkers to predict CO-induced cardiomyopathy.

Cardiac markers

Total

(N = 850)

No cardiomyopathy

(N = 749)

Cardiomyopathy

(N = 101)

P

Initial CK-MB

(ng/mL)

2.0 (0.7–9.8)

1.55 (0.60–4.95)

25.60 (9.20–60.30)

< 0.01

Peak CK-MB

(ng/mL)

2.5 (0.8–13.3)

1.80 (0.65–5.70)

31.80 (13.80–73.60)

< 0.01

Initial hsTnI

(ng/mL)

0.03 (0.01–0.27)

0.01 (0.01–0.12)

1.31 (0.38–3.16)

< 0.01

Peak hsTnI

(ng/mL)

0.05 (0.01–0.50)

0.02 (0.01–0.23)

1.84 (0.58–4.90)

< 0.01

BNP

(pg/mL)

15.00 (5.00–41.50)

15.00 (5.00–41.50)

89.00 (29.00–255.00)

< 0.01

Abbreviations: CO = carbon monoxide; CK-MB = creatinine kinase-myocardial band; hsTnI = high-sensitivity troponin I; BNP = brain natriuretic peptide.

Discussion: “Both CK-MB and hsTnI concentrations at initial presentation, but not BNP concentration, were associated with CO-induced TTE-determined myocardial dysfunction.” Although the performance of BNP may not be as good as other markers in the present study, BNP levels were greater in Cardiomyopathy group than the No cardiomyopathy group (Table 2; P<0.01). Please consider rewording this statement.

Response> We accepted your opinion and rewrote for clarifying our findings.

“All CK-MB, hsTnI, and BNP concentrations at initial presentation were associated with CO-induced TTE-determined myocardial dysfunction. Diagnostic abilities of CK-MB and hsTnI levels were better than that of BNP.”

Discussion: “Although serial measurements of CK-MB and hsTnI concentrations identified their peak levels, these peak concentrations were not superior to concentrations at admission for predicting CO-induced cardiomyopathy.” While statement may be true, it is encouraged to add the comparison of AUCs to the results demonstrating statistical significance.

Response> We performed additional analysis for the comparisons of AUCs by using methodology of Delong et al. between initial and peak levels of CK-MB and hsTnI concentration and mentioned in the results section.

Initial CK-MB and peak CK-MB: difference between areas 0.003, p value = 0.20

Initial hsTnI and peak hsTnI: difference between areas 0.004, p value = 0.71

"Comparison of AUC between initial and peak concentrations of each biomarker showed no statistical differences (difference between areas 0.003; p = 0.20 for initial vs. peak CK-MB, difference between areas 0.004; p = 0.71 for initial vs. peak hsTnI)"

Whether CO-induced cardiomyopathy is a clinical entity has yet to be established. This should be addressed by discussing the current knowledge from preliminary studies.

Response> Thank you for your suggestions. We reviewed previous reports about CO-induced cardiomyopathy and added the sentences in the Discussion section.

“The heart is the primary target of acute CO poisoning via direct toxic effect or secondary to ischemic hypoxia [2,3]. Previous reports announced that CO caused various cardiovascular complications, such as arrhythmia, heart failure, and myocardial infarction [17]. Although its exact epidemiology and pathophysiology have not been fully discovered, recent studies tried to evaluate the risk factors, patterns, clinical outcomes, and prognosis [18,19]. Jung et al. reported that CO-induced cardiomyopathy had similar characteristics and pattern with stress-induced cardiomyopathy with favorable outcomes. Moreover, they suggested that myocardial stunning caused by a catecholamine surge plays a key mechanism in the occurrence of cardiomyopathy [19].”

Discussion: “Because the concentrations of these enzymes increase at different rates, we evaluated whether combinations of their cutoff values could enhance their diagnostic abilities, including sensitivity and specificity. However, none of these combinations improved outcomes.” Relevant data should be reported in the Results section.

Response> We agreed with your opinion and added the results of the combinations of cutoff values in table 4.

“Meanwhile, combinations of the cutoff values could not improve diagnostic performances.”

Table 4. Performance parameters for predictors of CO-induced cardiomyopathy.

Variables

Sensitivity (%)

Specificity (%)

PPV (%)

NPV (%)

PLR

NLR

Initial CK-MB > 1.5 ng/mL

95.1

44.0

32.0

97.0

1.7

1.1

Peak CK-MB > 1.9 ng/mL

97.0

46.2

33.3

98.3

1.8

0.1

Initial hsTnI > 0.01 ng/mL

96.0

41.8

31.4

97.4

1.7

0.1

Peak hsTnI > 0.03 ng/mL

95.2

44.0

32.5

97.0

1.7

0.1

BNP > 14.5 pg/mL

87.7

37.9

33.5

89.6

1.4

0.3

Initial CK-MB > 1.5 + hsTnI > 0.01

96.0

31.0

27.9

96.6

1.4

0.1

Peak CK-MB > 1.9 + hsTnI > 0.03

97.0

36.3

29.7

97.8

1.5

0.1

Abbreviations: CO = carbon monoxide; PPV = positive predictive value; NPV = negative predictive value; PLR = positive likelihood ratio; NLR = negative likelihood ratio; CK-MB = creatinine kinase-myocardial band; hsTnI = high-sensitivity troponin I; BNP = brain natriuretic peptide.

Round 2

Reviewer 1 Report

The author well answered the questions and provided the suitably responses. I suggested the response about this study did not provide clinical outcomes and should be listed in limitation.

Author Response

The author well answered the questions and provided the suitably responses. I suggested the response about this study did not provide clinical outcomes and should be listed in limitation.

Response> Thank you for your generous comments. We mentioned the limitation of our study in the Discussion section.

“Finally, this registry did not include data about clinical outcomes, such as recovery rate, mortality, and neurologic deficit, and could not provide clinical impacts of CO-induced cardiomyopathy. A well-designed prospective study will be needed to reveal these questions.”

Reviewer 2 Report

The revised manuscript has been improved by correction and clarification.  The work is well-written and advances the understanding of CO-induced cardiovascular injury.

Minor comment:

The findings from Supplement Table 1 to 3 are worth mentioning, at least in brief, in the Results section 3.4.

Author Response

The revised manuscript has been improved by correction and clarification.  The work is well-written and advances the understanding of CO-induced cardiovascular injury.

Minor comment:

The findings from Supplement Table 1 to 3 are worth mentioning, at least in brief, in the Results section 3.4.

Response> Thank you for your generous comments. We added sentences about Supplement Table 1 to 3 in the Results section 3.4.

“Moreover, all initial and peak concentrations of CK-MB, hsTnI, and BNP were significantly higher in each type of dysfunction, including LV, RV dysfunction, and abnormal wall motions (Supplement Table 1–3).”
